# CRISPR/Cas-Mediated Resistance against Viruses in Plants

**DOI:** 10.3390/ijms23042303

**Published:** 2022-02-19

**Authors:** Zainul A. Khan, Rohit Kumar, Indranil Dasgupta

**Affiliations:** Department of Plant Molecular Biology, University of Delhi South Campus, New Delhi 110021, India; zainulbiotechnology@gmail.com (Z.A.K.); rt93kumar@gmail.com (R.K.)

**Keywords:** CRISPR/Cas, genome editing, plant viruses, multiplexing, virus resistance

## Abstract

CRISPR/Cas9 provides a robust and widely adaptable system with enormous potential for genome editing directed towards generating useful products. It has been used extensively to generate resistance against viruses infecting plants with more effective and prolonged efficiency as compared with previous antiviral approaches, thus holding promise to alleviate crop losses. In this review, we have discussed the reports of CRISPR/Cas-based virus resistance strategies against plant viruses. These strategies include approaches targeting single or multiple genes (or non-coding region) in the viral genome and targeting host factors essential for virus propagation. In addition, the utilization of base editing has been discussed to generate transgene-free plants resistant to viruses. This review also compares the efficiencies of these approaches. Finally, we discuss combinatorial approaches, including multiplexing, to increase editing efficiency and bypass the generation of escape mutants.

## 1. Introduction

Management of plant viral diseases is a matter of vital importance and concern to farmers. Viral diseases in crops cause enormous losses throughout the world. Both conventional and non-conventional approaches have been applied to develop plants resistant to viruses. The approach was applied for the development of tobacco and pepper tobamovirus resistance in plants in the 1930s [1,2]. Effector-triggered immunity, a mechanism to develop resistance to viruses, has been characterized in many plant species, and the resistance is mediated by the plant genome encoded *R*-genes [3,4]. The concept of pathogen-derived resistance (PDR) [5] was first applied to develop resistant tobacco plants against the tobacco mosaic virus (TMV) by the transformation and expression of the viral coat protein (CP) [6], followed by the application of this strategy to other plants [7,8,9,10]. Recent studies have shown that resistance can also be derived by RNA silencing, also known as RNA interference (RNAi), which plays an extensive role in antiviral defense in plants. RNAi is activated in the presence of double-stranded RNAs (dsRNAs), which leads to either the inhibition or suppression of genes [11,12]. Many approaches have been used to develop virus-resistant transgenic plants, including sense/antisense RNA, hairpin RNA, microRNA (miRNA) and artificial miRNA precursors [13,14]. Till now, more than 60 economically important plant virus species have been successfully targeted using RNAi technology [15]. Approaches using conventional breeding and transgenics such as PDR and RNAi have certain limitations. The breeding approach is a laborious and time-consuming process. The application of modern biotechnology and the development of genetic engineering have significantly accelerated the efficiency of breeding by modifying the host’s genetic information [16]. Transgenic products have to undergo regulatory approvals and deal with public acceptance, which are often time consuming. More recently, the application of naked dsRNA has also been used to trigger RNAi against viruses to circumvent some of the above issues. However, the window for protection against viruses using this approach is rather short, only 5 days post application [15,17].

As plant viruses evolve rapidly by recombination and mutation, the conventional strategy may fail to control these viruses. A durable resistance is required to overcome these challenges. The CRISPR (clustered regularly interspaced palindromic repeats)/CRISPR-associated 9 (CRISPR/Cas9) system has recently emerged as an efficient genome-editing tool for many eukaryotic species, including plants. This technology has some advantages over artificial microRNAs and RNAi for engineering virus resistance in plants by the disruption of essential viral or host gene instead of silencing those genes at the RNA level. Several studies have demonstrated the advantages of the CRISPR/Cas9 system to effectively confer resistance to DNA and RNA viruses in plants, either by directly targeting and cleaving the viral genome or by modifying the host genes [18,19,20,21,22,23,24,25,26,27,28,29,30,31,32,33,34,35,36,37] (Table 1 and Figure 1). Since the CRISPR-Cas system has a prokaryotic origin, the chances of plant viruses evolving mechanisms to antagonize it are unlikely, in contrast to RNAi, which is a plant-derived process and against which most viruses already carry suppressors [15,38]. Multi-targeting guide RNAs (gRNAs) is an efficient way to remove mutant viruses (escape mutants) developed upon CRISPR/Cas9-mediated genome editing [29,39].

## 2. CRISPR/Cas Nucleases

CRISPR/Cas systems, a site-directed genome-editing tool, administer adaptive immunity to archaea and bacteria against viruses and plasmids by using CRISPR RNAs (crRNAs), which direct Cas endonuclease to cleave invading nucleic acids on the basis of sequence complementarity [40]. The CRISPR/Cas system requires a Cas endonuclease and a customized single guide RNA (sgRNA) complimentary to target DNA. CRSIPR/Cas9 and other CRISPR/Cas systems are applied for targeted mutagenesis, multiplex genome editing, epigenetic modifications, gene replacement and gene knock-in and diagnosis of pathogens [41,42].

## 3. CRISPR-Mediated Immunity against Plant Viruses

On the basis of genomes, plant viruses are differentiated into six major groups: single stranded (ss) DNA viruses, double stranded (ds) DNA viruses, reverse-transcribing viruses, positive sense ssRNA viruses, negative sense ssRNA viruses and dsRNA viruses [43,44]. Plant viruses can either be targeted directly by CRISPR for dsDNA/ssRNA breaks or essential host component genes can be targeted for mutations so that the virus can no longer infect (Figure 1).

### 3.1. CRISPR-Mediated Immunity against DNA Viruses

The most common DNA viruses infecting plants belong to the family *Geminiviridae* and infect economically important crops such as cassava, cotton, cucurbits, mung beans, potato, soybean, tobacco, tomato, etc., in the tropical and sub-tropical regions of the world. Geminiviruses contain either monopartite or bipartite single-stranded (ss) circular DNA genome of 2.5 to 3.0 kb, encapsidated into twinned geminate particles. On the basis of genome organization, host range and virus vector, the family *Geminiviridae* is differentiated into 14 genera *viz. Becurtovirus*, *Begomovirus*, *Capulavirus*, *Citlodavirus*, *Curtovirus*, *Eragrovirus*, *Grablovirus*, *Maldovirus*, *Mastrevirus*, *Mulcrilevirus*, *Opunvirus*, *Topilevirus*, *Topocuvirus* and *Turncurtovirus*, among them *Begomovirus* being the largest, having more than 400 species [45,46,47,48]. Begomoviruses are transmitted by whiteflies belonging to the cryptic species *Bemisia tabaci*. Bipartite begomoviruses possess two genomic components, DNA A and DNA B. DNA A encodes six or seven open reading frames (ORFs), four or five ORFs (AC1, AC2, AC3, AC4 and AC5) are in complimentary sense, while two ORFs (AV1 and AV2) are in virion sense orientations. DNA B contains only two ORFs BV1 (virion sense) and BC1 (complimentary sense). The single genomic component of monopartite begomoviruses resembles DNA A of bipartite begomoviruses. A number of half-size (betasatellite and alphasatellite) and quarter-size (deltasatellite) ssDNA molecules are also found in many geminivirus-infected plants, and these have been named satellites. Betasatellites (genus: *Betasatellite* and family: *Tolecusatellitidae*) are responsible for symptom induction/severity in plants and depend on the helper virus for their replication, encapsidation and systemic movements. The alphasatellites are able to replicate autonomously and depend upon the helper virus for their movement within plants and whitefly transmission between plants. Alphasattelites regulate the virulence of the betasatellite-begomovirus complex [49,50]. The deltasatellites are replicated and accumulated in the inoculated plants only in the presence of the helper virus and are not known to have any function [51].

Mastreviruses have a monopartite genome of 2.6–2.8 kb and are transmitted by leafhoppers (*Psammotettix alienus*). The mastrevirus genome has four ORFs, two in virion sense (CP/V1 and MP/V2) and two in complimentary sense orientation (RepA/C1 and Rep/C1:C2) [52].

The genome of curtoviruses consists of a single DNA component of 2.9–3.0 kb, encoding seven ORFs, four in complimentary sense (C1/Rep, C2/TrAP, C3/REn and C4) and three in virion sense (V1/CP, V2/MP and V3) and are transmitted by leafhoppers [53].

The DNA of geminiviruses is replicated as double-stranded (ds) replication intermediates in the nucleus of infected plants, and thus have been targeted for mutagenesis using CRISPR/Cas by a number of researchers (Figure 1). The first reports for such an approach were demonstrated in plants commonly used in the laboratory such as *Nicotiana benthamiana* and *Arabidopsis thaliana* [18,19,20] (Table 1).

Baltes et al. (2015) designed sgRNAs which target both coding (Rep and RepA) and non-coding regions (LIR; large intergenic region) of bean yellow dwarf virus (BeYDV, genus *Mastrevirus*) genome. First, they checked the activity of these sgRNAs transiently in *N. benthamiana* plants. The *CP* and movement protein genes of BeYDV were replaced with enhanced green fluorescent protein (eGFP) gene to check the activity of sgRNAs against BeYDV. The Cas9-sgRNA and eGFP-BeYDV constructs were co-infiltrated in *N. benthamiana* and average eGFP intensity was quantified at 5 days post inoculation (DPI), and the sgRNAs which showed significant reduction in eGFP intensity were selected. The two most active sgRNAs targeting BeYDV Rep binding site (gBRBS) in LIR and BeYDV M3 motif (BM3) of Rep/RepA that showed most insertions and deletions (INDELs) percentages in the target site of viral genome were further used for the development of transgenic *N. benthamiana*. Upon challenge virus inoculation, the transgenic *N. benthamiana* plants either expressing sgRNA-BRBS and Cas9 or sgRNA-BM3 and Cas9 showed 71% and 78% reduction in viral titers, respectively, and mild symptoms as compared with the plants expressing Cas9 alone.

Ji et al. (2015) utilized a similar approach targeting beet severe curly top virus (BSCTV) that belongs to the genus *Curtovirus*. Upon virus inoculation, the gRNA targeting intergenic region (IR), *CP* or *Rep* gene of BSCTV showed the reduction in the virus titer by 30–90%, 20–90% and 70–95%, respectively in transiently expressed sgRNA-Cas9 in *N. benthamiana* plants compared with wild type plants. Furthermore, transgenic *N. benthamiana* and *Arabidopsis* plants over-expressing sgRNA-Cas9 targeting IR and *Rep* gene, respectively showed significant inhibition of BSCTV replication, which was correlated with the Cas9 expression level. Transgenic *N. benthamiana* line with lower level of Cas9 expression showed mild symptoms, while the transgenic line with higher levels exhibited no symptoms. Virus inhibition was also examined in T2 transgenic *Arabidopsis* lines, while non-transgenic *Arabidopsis* plants showed severe symptoms such as leaf curling, deformed floral structures and curled and stunted inflorescences.

Ali et al. (2015) targeted both coding and non-coding regions of the DNA of tomato yellow leaf curl virus (TYLCV; genus *Begomovirus*) using a CRISPR/Cas9 approach into *N. benthamiana* plants. The gRNAs targeting Rep, CP and IR were individually cloned into tobacco rattle virus (TRV) RNA2 genome and agroinfiltrated into Cas9 overexpressing *N. benthamiana* (NB-Cas9OE). Furthermore, they amplified the targeted region of the virus sequence containing IR (560 bp), followed by cloning and sequencing. Out of 300 sequenced clones, 42% of clones showed targeted modifications within IR sequence. The results show that this approach efficiently targeted TYLCV and introduced mutations. The sgRNAs targeting stem-loop sequences (origin of replication) within the IR of TYLCV were the most effective. About 85% of *N. benthamiana* plants expressing IR-sgRNA did not show TYLCV symptoms, only 15% of plants showed mild symptoms, while 73.3% and 88.7% of *N. benthamiana* plants having CP-sgRNA and Rep-sgRNA constructs, respectively showed mild symptoms.

The stem-loop sequence of the origin of replication in the IR is conserved in all geminiviruses. Ali et al. (2015) investigated the possibility of targeting different viruses with a single sgRNA. They designed an IR-sgRNA that contains the invariant TAATATTAC sequence common to all geminiviruses and tested this IR-sgRNA against TYLCV, beat curly top virus (BCTV, genus *Curtovirus*) and Merremia mosaic virus (MeMV, genus *Begomovirus*). The results show that a single sgRNA was able to target multiple viruses (TYLCV, BCTV, MeMV), all of them having the invariant TAATATTAC sequence. The NB-Cas9OE plants infiltrated with a single TRV vector containing two sgRNA targeting CP and IR showed greater reduction in TYLCV titer and recovery from disease symptoms. About 95% of plants did not display viral symptoms and 11% showed presence of viral DNA [20].

Furthermore, Ali et al. (2016) designed the sgRNA to target IR, CP and Rolling Circle Rep II (RCRII) domain of Rep of cotton leaf curl Kokhran virus (CLCuKoV; genus *Begomovirus*), MeMV and tomato yellow leaf curl Shahdadpur virus (TYLCSV; genus *Begomovirus*) using the CRISPR/Cas9 approach. They used TRV-mediated delivery of individual sgRNA constructs into *N. benthamiana* Cas9-OE plants through the transient infiltration method and checked the inhibition of viral accumulation upon CLCuKoV, MeMV and TYLCSV inoculation. Mutation analysis shows that high levels of indel were detected at both CP (18–49%) and RCRII of Rep (35–45%) of CLCuKoV. They could not detect mutations in the IR of CLCuKoV targeted plants. Furthermore, they investigated whether the CLCuKoV could be targeted by invariant sgRNAs (sgRNA of IR of BCTV, TYLCV and MeMV against CLCKoV), but the mutation analysis did not reveal significant levels of indel formation. In contrast to CLCuKoV, the sgRNA targeting IR of MeMV showed mutation (13–19%) in the IR of MeMV. Similar to the IR, targeting the CP and RCRII of Rep of MeMV produced high rates of indel generation. IRs interact with a specific Rep as they are strain specific, so they used a different strain of TYLCV such as TYLCV2.3 and TYLCSV. They designed TYLCV2.3-IR-sgRNA and TYLCSV-IR-sgRNA targeting TYLCV2.3 and TYLCSV, respectively. They could not detect indels in TYLCSV-IR-sgRNA targeted TYLCSV, while they detected the mutation when they used TYLCSV-sgRNA or TYLCV2.3-sgRNA to target TYLCV2.3. Indels were detected in the CP (71%) and RCRII of the Rep (41%) of TYLCSV, in contrast to the inability to detect indels in the IRs.

The CRISPR/Cas9 approach creates a selection pressure on the viruses to evolve quickly that can result in the generation of mutants, some of which may become resistant to editing. In addition, the host DNA repair machinery could be used by the viruses for the generation of variants capable of escaping the CRISPR/Cas9 machinery [22]. The sgRNAs targeting noncoding IRs of CLCuKoV and TYLCSV were capable of interference but unable to generate virus variants, while those targeting the coding regions of all the viruses were efficient for interference and generating virus variants [21]. Furthermore, Ali et al., 2016 investigated the replication ability and systemic movement of mutated viral variants that could evade the CRISPR/Cas9 system. Mutations in coding sequences of the viruses were replicated and could move systemically, thereby evading the CRISPR/Cas9 machinery, while IR mutants failed to replicate and move systemically [21].

*N. benthamiana* and tomato (*Solanum lycopersicum*) plants displaying resistance against TYLCV were generated [22] through the CRISPR/Ca9 approach. In this study, they designed sgRNAs targeting CP, Rep and IR of TYLCV. Targeted genome-editing-based resistance remained active across multiple generations in tomato and *N. benthamiana* plants. Transgenic *N. benthamiana* plants expressing Cas9 and sgRNA for CP, Rep and IR showed indel mutations up to 31%, 46% and 22%, respectively upon TYLCV inoculation. As the silencing of the T-DNA inserted genes (transgene) happens in plants [54], Tashkandi et al. (2018) tested virus interference through multiple generations. T2 transgenic tomato lines expressing sgRNA for CP and Rep genes of TYLCV exhibited 34–69% and 14–37% indels, respectively in the viral DNA. The TYLCV accumulation decreased up to 69% and 76% in T2 tomato plants expressing CP and Rep sgRNA, respectively. TYLCV-inoculated T3 progeny expressing Cas9, CP and Rep sgRNAs revealed indels of 49% and 29%, respectively, and exhibited low accumulation of viral genome as compared with WT plants. Furthermore, they checked whether CRISPR/Cas9 machinery developed an escape variant of TYLCV or not. Sap was collected from TYLCV-infected transgenic *N. benthamiana* plants expressing Cas9 and sgRNA targeting Rep of TYLCV and followed by inoculation of TYLCV into *N. benthamiana* wild-type plants. Sequence analysis confirmed accumulation of Rep sequence variants of TYLCV genome and further analysis demonstrated that all the variants have functional and contiguous ORFs with diverse amino acids at targeted Rep region, thereby allowing virus replication and systemic movement. Since any mutation in the seed region (9–12 nt) of the spacer sequence near the PAM would eliminate or limit the efficiency (ability) of the CRISPR/Cas9 system to cause mutation in the target, the generation of editing-resistant virus variants becomes likely [55].

In a recent study, dual sgRNAs targeting two essential regions of IR and Rep (C1) of Cotton leaf curl Multan virus (CLCuMuV, genus *Begomovirus*) genome was utilized [23]. A single construct having Cas9, gRNA-IR and gRNA-Rep was used to generate transgenic *N. benthamiana* plants, which showed complete resistance against CLCuMuV and no accumulation of viral DNA compared with the WT plants and the plants expressing only Cas9.

In another study, four sgRNAs targeting overlapping regions of MP and CP, Rep/RepA at N-terminus of protein, LIR and C-terminus of Rep of Wheat dwarf virus (WDV, genus *Mastrevirus*, family *Geminiviridae*) were designed [24]. All four sgRNA and Cas9 were cloned into a single binary vector (WDVGuide4Guard) and used to transform barley plants. The viruliferous leafhoppers were inoculated onto the transgenic plants to mimic the natural infection process. At 7 DPI, both the transgenic and control plants revealed the presence of virus specific DNA. However, at 112 DPI, three out of four transgenic lines neither showed any symptoms nor the accumulation of any viral DNA, while the control plants showed typical symptoms. Next, they investigated the presence of different sgRNA in virus-inoculated transgenic plants at 112 DPI. They found the expression of all the sgRNA except sgRNA targeting IR. Interestingly, mutation analysis revealed that only the sgRNA targeting Rep/RepA at the N-terminus showed the presence of three-nucleotide insertion in the WDV genome.

Recently, monopartite begomovirus (cotton leaf curl virus; CLCuV) and associated betasatellite DNA in *N. benthamiana* plants were targeted for editing [25]. Targeting Rep of CLCuV and βC1 of betasatellite DNA by multiplex approach showed lower virus titers and delay in symptom development when compared with the plants inoculated only with sgRNA targeting Rep. They designed two constructs: in the first construct they cloned sgRNA targeting Rep of CLCuV, while in the second construct they cloned two sgRNAs targeting Rep and βC1 of betasatellite DNA. These two constructs containing sgRNAs and Cas9 were agroinfiltrated in *N. benthamiana* leaves. After, 2 DPI infectious clones of CLCuKoV and Cotton leaf curl Multan betasatellite were agroinfiltrated in the sgRNA inoculated *N. benthamiana* plants. *N. benthamiana* plants expressing Cas9 and sgRNA targeting Rep of CLCuV showed 40–70% reduction in virus titer compared with the control plants. While the multiplex approach revealed the reduction in virus accumulation by 60–80% and 3–6 days’ delay in symptom development compared with the control plants.

Mehta et al. (2019) designed six sgRNA targeting AC2/AC3, AC1/AC2 and AV1/AV2 overlapping region, AC1, AV1 and the IR region of bipartite begomovirus, the African cassava mosaic virus (ACMV). To check the effectiveness of six different sgRNAs against the viral template, they performed in vitro cleavage assay. They selected sgRNA targeting the AC2/AC3 region of ACMV for stable expression in host cassava plants because this sgRNA performed well and has best predicted efficiency during in vitro cleavage assay. Transgenic cassava lines expressing sgRNA, Cas9 and control cassava plants expressing only Cas9 and wild-type control plants did not show any significant differences in virus titer, disease incidence or symptoms severity. Sequence analysis of virus population in plants containing the CRISPR/Cas9 construct in three independent lines showed a single-nucleotide “T” insertion in the AC2 ORF that resulted in H54Q substitution and produced a premature stop codon and reduced the AC2 protein from 136 to 62 amino acid. However, the mutation resulted in the generation of a new ORF, which led to the formation of two distinct translational units for AC2. The authors generated an infectious clone of the mutant virus to check whether it replicated independently in *N. benthamiana* plant or was dependent upon the wild-type (WT) ACMV for its replication and accumulation. They observed that while the mutant ACMV failed to accumulate independently, it was viable when co-inoculated with WT-ACMV. Thus, they concluded that the CRISPR/Cas9 approach can give rise to the rapid evolution of editing-resistant viruses.

In addition, CRISPR/Cas machinery has been applied to pararetroviruses, which have ds DNA genome, such as Cauliflower mosaic virus (CaMV, genus *Caulimovirus*, family *Caulimoviridae*) and endogenous banana streak virus (eBSV, genus *Badnavirus*, family *Caulimoviridae*). Pararetroviruses have a circular DNA genome and replicate through transcription/reverse transcription [56]. In spite of their non-integrative replication cycle, a number of integrated sequences known as endogenous plant pararetroviruses of family *Caulimoviridae* have been reported in both monocots and dicots [57,58,59,60]. Liu et al. (2018) designed six individual sgRNAs targeting distinct sites in the CP of CaMV. These six sgRNAs were cloned as a linear array into the binary vector containing Cas9 and transformed into *Arabidopsis* plants. About 10–15% of transgenic *Arabidopsis* plants expressing sgRNAs and Cas9 showed symptoms after CaMV inoculation, while 95% wt and 85% transgenic *Arabidopsis* plants expressing only sgRNAs without Cas9 exhibited symptoms. Symptomatic transgenic lines showed a 20–52% reduction in viral titer, while the viral DNA was undetectable in the symptomless transgenic lines. Furthermore, they tested editing events at earlier and later stages of viral infections. At 3 DPI, 43% of clones (56/129 clones) had shorter CP fragments. Eight randomly picked clones (of 56 clones) were sequenced. These clones showed deletions in CP sequences, ranging from 590 bp to 1351 bp.

Endogenous pararetroviruses (EPRV) belong to the family *Caulimoviridae,* and EPRV has been identified in the nuclear genomes of a wide range of Angiosperms [61]. Endogenous viral elements (EVE) are incorporated into the host chromosome of germ cells for the vertical transmission and potential fixation in the host population and play an important role in their hosts’ evolution [62]. All types of eukaryotic viruses can produce/generate EVEs [62]. EPRVs are the most abundant known plant virus integrations [61]. CRISPR/Cas9 technology was also applied to mutate the eBSV genome in the host plantain cv. Gonja Manjaya [28]. Most of the edited lines (75%) tested remained asymptomatic as compared with the non-edited control plants under water stress conditions, which confirmed the inhibition of conversion of eBSV into infectious virus particles [28].

A durable and transgene-free virus-resistant plant can be developed by targeting susceptibility factors (S-gene) that interact with the viral proteins during infection and are necessary for symptom development (virus propagation). In the recent work by Pramanik et al., 2021, the S-gene named *SlPelo* in tomato (*S. lycopersicum*) was targeted to accomplish TYLCV resistance. The *SlPelo* gene synthesizes an mRNA surveillance factor called as Pelota (PELO), encoded by Ty-5 locus [63]. The function of PELO protein is in ribosome recycling during translation, and CRISPR/Cas9-mediated *SlPelo* knockout mutant tomato plants showed a reduction in TYLCV accumulation [30,63]. Pramanik et al. (2021) designed a plasmid vector carrying sgRNAs for the editing of a single gene which increases the chance of large deletions and knockout generation as previously reported [64]. SlPelo protein carries three conserved eukaryotic translation termination factor 1 (eRF1) domains named eRF1_1, eRF1_2 and eRF1_3. These domains are vital for the functioning of ribosome cycling followed protein synthesis during translation. They demonstrated that indel mutations in the coding region of eRF1_1 led to the production of dysfunctional PELO protein. Nine sgRNAs, each targeting a different region of *SlPelo* coding sequence, were designed and cloned in multiplex CRISPR vectors with each vector carrying sgRNAs against two or three regions of coding sequence. They successfully transformed commercial tomato BN-86 line. Screening of transformed G0 (genome-edited generation 0) lines by *SlPelo* locus sequencing reported lower efficiency of sgRNAs, which might be due to limited availability of Cas9 in a multiplexed system or non-accessibility of target sites to sgRNA due to chromatin confirmation. One sgRNA showed successful editing at target site in the *SlPelo* locus, and these *SlPelo* mutant G0 plants were found to be biallelic or mosaic, with low indel frequencies (>10%) in coding exon. Nevertheless, screening of segregating G1 generation plants successfully produced biallelic knockout mutants.

In earlier reports, RNAi-mediated silencing of *SlPelo* was successful in conferring TYLCV-resistant phenotype and decreased viral titers were observed [63]. Similarly, when Pramanik et al. (2021) challenged CRISPR based *SlPelo*-edited knockout G1 (genome-edited generation 1) plants against TYLCV, they reported no symptoms in TYLCV-infected plants and a significant reduction in viral load. Though there were few plants with mild symptoms, the authors explained this might be due to higher inoculation load or partial loss of function in PELO. This study demonstrated the editing of S-genes as an effective approach to engineer the viral resistance in tomato crop.

### 3.2. CRISPR-Mediated Immunity against RNA Viruses

A large number of viruses infecting economically important crops belong to the category of RNA viruses [65,66]. CRISPR/Cas technology has been applied successfully to develop resistant plants against RNA viruses (Table 1). As discussed in the above section, DNA viruses, viral DNA and S-gene were targeted, similarly in the case of RNA viruses, both viral RNA and host factors, which interact with viral proteins during infection, have been targeted for editing (Figure 1). Translation in eukaryotes requires a large number of translation initiation factors (eIFs). One of the important eIFs required is the eIF4E, which interacts with 5′ m7G cap structure of eukaryotic mRNAs. Translation of eukaryotic mRNAs is also enhanced by the association of polyA-binding protein (PABP) with the 3′ polyA tail and binding of a large scaffold protein eIF4G to both PABP and EIF4E, leading to the circularization of mRNA. A core complex of eIF4F is formed by the tight association between eIF4E and eIF4G. eIF4G also interacts with DEAD-(Asp-Glu-Ala-Asp)-box ATPase, eIF4A and ATP-dependent RNA helicase [67]. eIF4A unwinds the mRNA to help ribosome scanning. The key translational factors eIF4E, eIF4G, eIF4A and DEAD-box RNA helicase are used by viruses for the translation of their mRNAs and are attractive targets for antiviral strategies. The protein complex eIF3 interacts with both 40S ribosome subunit and eIF4G, thereby bringing the 43S pre-initiation complex, which includes the 40S subunit and ternary complex eIF2-GTP-tRNA^Met^, to the mRNA [67]. Gene duplication in higher plants results in the formation of a second eIF4F complex, known as eIF(iso)4F, which consists of eIF(iso)4E and eIF(iso)4G [68,69]. *A. thaliana* encodes three isoforms of the eIF4F/eIF(iso)4F large subunits, eIF4G, eIF(iso)4G1 and eIF(iso)4G2 and four isoforms of the small subunit cap-binding protein eIF4E1, eIF4E2, eIF4E3 and eIF(iso)4E. eIF4E1 and eIF(iso)4E share 41% amino acid sequence identity and both have a similar molecular mass of approximately 24 kDa. However, eIF(iso)4G isoforms differ in molecular mass from the eIF4G (83/86 kDa vs. 168 kDa, respectively) and have 27% amino acid identity to eIF4G [70]. Their C-terminal domains share common binding motifs for eIF4E, eIF4A and eIF3, while the N-terminal region is truncated in eIF(iso)4G. In *Arabidopsis*, the knock-out of eIF(iso)4G results in a severe defect in plant health [70]. In contrast, eIF4E or eIFiso4E knock-out or when the gene is down-regulated show little signs of impairment [71,72]. Mutation in eIF(iso)4E is compensated by the increased expression of eIF4E, indicating functional redundancy between two isoforms [67,71].

Interestingly, potyviruses can use either eIF4F or its isoform eIF(iso)4F for translating viral RNA. The knock-out of one isoform of eIF4G or eIF4E can result in the development of plants resistant to potyvirus without affecting the plant health and making them an appealing source of engineered or natural resistance [67,69].

The uncapped RNAs of potyviruses and secoviruses are covalently linked with viral protein genome linked (VPg) at 5′ end [73]. Potyvirus VPgs interact with either eIF4E or eIF(iso)4E isoforms, while the cellular mRNAs utilize both the isoforms. The essentiality of this interaction has been shown between VPg of Turnip mosaic virus (TuMV) and host eIF(iso)4E [74] (Figure 1).

Previously, eIF4E and its paralogue eIF(iso)4E have been reported as recessive resistance alleles against various potyviruses in different hosts. Earlier reports showed that mutations in eIF(iso)4E, which are required for viral replication in *Arabidopsis*, caused by either ethyl methanesulfonate or transposon insertion, resulted into complete resistance to several potyviruses including TuMV, Tobacco etch virus and Lettuce mosaic virus [71,75].

TuMV belongs to the family *Potyviridae* of genus *Potyvirus,* having a positive sense ssRNA genome of about 10 kb. CRISPR/Cas9 technology was utilized [34] to target eIF(iso)4E of *A. thaliana* for the generation of TuMV resistance. T1 transgenic lines were generated, which expressed sgRNA against eIF(iso)4E, and Cas9 displayed mutations in the target gene. The investigators recovered 55 transgene-free lines, of which 70.9% of transgene-free (39 out of 55 plants) plants harbored mutations in eIF(iso)4E, 4 mutations being homozygous. T3 homozygous mutant lines exhibited complete resistance against TuMV, no differences in dry weights and flowering times under standard growth conditions when compared with wild-type plants.

Chandrasekaran et al. (2016) designed sgRNAs to target 5′ and 3′ termini of the *eIF4E* gene of cucumber (*Cucumis sativus* L). Two *eIF4E* genes, *eIF4E* (236 aa) and *eIF(iso)4E* (204 aa), shared 60% amino acid and 56% nucleotide homologies between them. They targeted two regions in the *eIF4E* of cucumber, which have no homology in the *eIF(iso)4E*. Five T_0_ transgenic cucumber lines were selected, three expressing sgRNA1 (5′ terminal of *eIF4E*) and two expressing sgRNA2 (3′ terminal of *eIF4E*). The plants expressing sgRNA1 showed mutations in eIF4E, while sgRNA2 expressing plants did not show genome editing in T0. Sequence analysis of the cloned target region of T_0_ transgenic plants having sgRNA1 showed two types of mutation, a 20 nucleotide deletion around the PAM and 1 nucleotide deletion at 3 bp upstream of PAM. T0 mutant plant (derived from “Ilan”, a multi-pistillate, parthenocarpic glasshouse cucumber) was cross-pollinated with “Bet Alfa” (a monoecious, non-parthenocarpic and open field cucumber) for propagation by seeds. As expected for a single transgene locus, the segregation of transgenic to non-transgenic in the T_1_ generation was observed approximately 1:1. Four types of plants appeared in T_1_ generation: a 20 nt deletion (non-transgenic, lacking Cas9), one nt deletion (transgenic), one nt deletion (non-transgenic, lacking of Cas9) and both one and 20 nt deletion (non-transgenic, lacking of Cas9). The non-transgenic T_1_ plants having both one nt and 20 nt deletion were cross-pollinated once again with the monoecious “Bet Alfa”. The T_2_ progeny representing hemizygous for one nucleotide deletion and 20 nt deletion were self-pollinated for the generation of T_3_ lines. In T_3_ population, 20 nt deletion segregated in Mendelian manner in 1:2:1 ratio (homozygous: heterozygous: wild type without mutation). T_3_ non-transgenic homozygous, heterozygous and wild-type (non-mutant) plants were inoculated with the Cucumber vein yellowing virus (CVYV, genus *Ipomovirus*), Zucchini yellow mosaic virus (ZYMV, genus *Potyvirus*), Papaya ring spot mosaic virus-W (PRSV-W, *Potyvirus*), Cucumber mosaic virus (CMV, genus *Cucumovirus*) and Cucumber green mottle mosaic (CGMMV, genus *Tobamovirus*). T_3_ homozygous mutant lines showed complete resistance against CVYV, ZYMV and PRSV-W and 100% disease incidence was recorded against CMV and CGMMV at 14 DPI. CMV and CGMMV have 5′ capped RNA. Uncapped viruses (potyviruses) require eIF4E and eIF(iso)4E host factors for their translation that have VPg protein covalently linked to the viral RNA 5′ [76]. All the non-homozygous (both heterozygous and non-mutant plants) plants showed symptoms against these viruses at 14 DPI.

CRISPR/FspCas technology was used to developed resistance in rice (*Oryza sativa* var. indica cv. IR64) against Rice tungro spherical virus (RTSV, Family *Secoviridae*) [36]. Rice production across tropical Asia is hampered by Rice tungro disease (RTD). RTD is caused by the interaction between Rice tungro bacilliform virus (RTBV), having a double-stranded DNA genome (family *Caulimoviridae*, genus *Tungrovirus*) and RTSV and having a single-stranded RNA genome (Family *Secoviridae*, Genus *Waikavirus*). Both RTBV and RTSV are transmitted by the green leafhopper (GLH) species. RTBV is responsible for symptom development, while RTSV helps in the transmission of RTBV by GLH and enhancing disease symptoms [77,78]. Macovei et al. (2018) designed sgRNA targeting eIF4G to develop RTSV-resistant rice. The presence of mutations was seen in 43 of 72 lines of T0 transgenic rice plants and a majority (62%, 27 of 43) were biallelic or chimeras (34.8%, 15 of 43). On further analysis of the type of mutations, 37.2% (16 of 43) of plants showed deletions, whereas 34.8% (15 of 43) of events exhibited a combination of substitutions and deletions and 6.9% (3 of 43) had a combination of insertions, deletions and substitutions, while just insertions or just substitutions were not detected in any transgenic events. The authors reported that the targeted mutagenesis was highly efficient, and the mutations were inherited in the next generation. Not all, but some mutant plants having in-frame mutation in SVLFPNLAGKS sequence adjacent to the YVV residues (amino acid sequence of target region of eIF4G) were resistant to RTSV. The most common mutation in the amino acid sequence of eIF4G of RTSV-resistant T2 plants were the NL residue within the SVLFPNLAGKS sequence. Other than YVV residues of SVLFPNLAGKSYVV sequence [79,80], NL residues are particularly important in determining the rice’s response to RTSV. SVLFPNLAGKS residues may play a role in the interaction between eIF4G and an RTSV protein, which influences rice’s response to RTSV. Off-target effects and Cas9 were not detected in the selected T_2_ plants. Agronomic parameters, in particular number of panicles and number of filled grains, plant height and total filled-grain weight, were quantified in the RTSV inoculated T2 plants and compared with RTSV inoculated wild-type IR64 plants. The number of filled grains and the total filled-grain weight were recorded to be almost double in the mutant T2 plants as compared with wild-type IR64 plants upon RTSV inoculation under controlled conditions (glasshouse conditions).

Recent reports showed the discovery of Cas endonucleases, namely FnCas9 from *Francisella novicida,* that target RNA molecules, which could be a new tool to target plant RNA virus genome. These endonucleases have been used in targeting the viral RNA genomes of plants [31,32,33,81,82]. Zhang et al. (2018) used CRISPR/Cas of FnCas9 to provide resistance against the RNA viruses, CMV and TMV in *N. benthamiana* and *Arabidopsis* plants. They designed sgRNA targeting only CMV, they further tested the efficacy of this sgRNA to target and inhibit the other RNA virus, TMV. FnCas9 and sgRNA infiltrating *N. benthamiana* plants at 5 DPI showed a 40–60% reduction in viral (CMV) RNA compared with the control vector-inoculated plants. At 14 DPI, infected vector control plants exhibited typical CMV symptoms (leaf shrinkage), while no severe leaf shrinkage symptoms were recorded in either systemic or inoculated leaves. Virus accumulation was 50–60% lower in sgRNA inoculated plants at 14 DPI. Similarly, *N. benthamiana* plants agroinfiltrated with sgRNA-FnCas9 targeting TMV RNA fused with GFP sequence (TMV-GFP) revealed up to 90% reduction in both GFP expression and TMV accumulation. Furthermore, they checked the inhibition of CMV in sgRNA-FnCas9 expressing transgenic *Arabidopsis* plants. T2 transgenic homozygous plants showed mild or no symptoms, while severe symptoms including leaf deformity and delayed growth were observed in control plants after two weeks of CMV inoculation. T2 and T6 transgenic plants showed a 70% to 85% reduction in viral accumulation compared with wild-type or mock vector-inoculated *Arabidopsis* plants. The resistance was inheritable, and the progenies of subsequent generations showed an 85% reduction in virus titer compared with wild type plants.

The functionality and characterization of Cas13a (previously known as C2c2 CRISPR/Cas system) from *Leptotrichia shahii* (LshCas13a) demonstrated that the single Cas13 protein is a programmable RNA-guided ssRNA ribonuclease and provides immunity to bacteriophage in the bacteria *Escherichia coli* [83]. Cas13 cuts the target RNA outside the crRNA binding site, presumably within exposed ssRNA loop regions, while Cas9 cleaves the target DNA within the crRNA heteroduplex at defined positions. Cleavage of target RNA as well as other nonspecific RNAs occurs upon the binding of the target RNA with the Cas13-crRNA complex. Cas13-crRNA complex-mediated cleavage of unrelated RNAs (collateral RNAs) takes place only in the presence of target RNA [83]. A recent work shows the use of Cas13a nuclease targeting ssRNA viruses TuMV in *N. benthamiana* [81], Potato virus Y (PVY) in potato [32], TMV in *N. benthamiana*, Southern rice black-streaked dwarf virus (SRBSDV) and Rice Stripe Mosaic Virus (RSMV) in rice [33] (Table 1).

PVY is considered the most devastating viral pathogen infecting potatoes. Zhan et al. (2019) utilized CRISPR/Cas13 to protect potato plants from PVY. They designed sgRNAs targeting P3 (membrane protein involved in pathogenicity, movement, systemic infection and virus replication), CI (forms the laminate cytoplasmic inclusion bodies, involved in infection and virus movement), NIB (RNA-dependent RNA polymerase (RdRp) that participates in the replication of the viral RNA) and CP (virion assembly, cell-to-cell and systemic movement and vector transmission) encoding regions of PVY RNA. The transgenic potato lines which shared the highest expression of Cas13a-sgRNA were selected for challenge with PVY. Cas13a-sgRNA, either targeting P3, CI, NIB or CP expressing transgenic potato plants showed more than 99% reduction in virus accumulation compared with WT (non-transformed control plants) potato plants. There were no disease symptoms in any of the edited plants, while the WT potato plants showed typical PVY mosaic symptoms at 25 DPI. The viral resistance was directly related to the expression level of Cas13a/sgRNA construct in transgenic potato. Furthermore, they challenged transgenic plants expressing CP-sgRNA with other strains of PVY (PVY^N^ and recombinant PVY^N:O^). More than a 90% reduction in virus titer was observed in transgenic plants. Next, they checked whether the PVY-resistant transgenic plants showed resistance to other unrelated potyviruses such as potato virus A and potato virus S that have low sequence similarity with PVY. When comparing respective four sgRNA sequences with the genomic DNA of PVA and PVS, spacer sequences were found to be less than 35% similar with their respective targets. They found that transgenic potato expressing CP-sgRNA plants and wild type did not show any significant difference in the virus accumulation.

Zhang et al. (2019) applied similar approaches to develop resistance in *N. benthamiana* towards TMV and in monocot plants (rice) against SRBSDV and RSMV. SRBSDV belongs to the genus *Fijivirus* of family *Reoviridae* and has a genome of 10 double-stranded RNA segments. SRBSDV infects rice in East Asian countries, such as China, Vietnam and Japan [84,85]. RSMV, an enveloped virus containing a single negative-sense RNA genome, is categorized under genus *Cytorhabdovirus* of family *Rhabdoviridae* [86]. First, the authors targeted TMV-GFP transiently in *N. benthamiana* plants using the CRISPR-Cas13a approach. A 70–80% reduction in TMV titer and correspondingly low expression of GFP were observed in the sgRNA-Cas13a-infiltrated plants compared with wild-type and control vector-inoculated plants. Furthermore, they targeted rice infecting viruses by designing the sgRNAs against SRBSDV and RSMV. T1 transgenic rice plants expressing sgRNAs-Cas13a showed mild symptoms and 60–80% less SRBSDV titer at 40 DPI as compared with the control vector-transformed or wild-type plants. The transgenic rice plants harboring Cas13a and sgRNA targeting RSMV showed low (70–80% reduction) virus accumulation and mild symptoms, while the wild-type plants and control vector-transformed plants showed typical symptoms including excessive tillering and slight dwarfing with leaves showing yellow stripes.

Furthermore, Mahas et al. (2019) used different variants of Cas13 such as CasRx, LwaCas13a and PspCas13b to check the efficacy and interference activities against single and multiple RNA viruses in *N. benthamiana*. *N. benthamiana* plants over-expressing CasRx (Cas13d from *Ruminococcus flavefaciens* XPD3002) showed efficient resistance against TuMV compared with transgenic plants over-expressing other Cas nucleases such as LwaCas13a (Cas13a from *Leptotrichia wadei*) and PspCas13b (Cas13b from *Prevotella* sp. P5-125). CRISPR-CasRx machinery was robust and efficient, either targeting a single RNA virus alone or two RNA viruses, and it also showed strong specificity against the targeted virus and did not exhibit collateral activity (cleavage of unrelated RNAs) in planta. The specificity of effector CasRX is essential for interference against viruses. Therefore, the specificity of CasRx against TMV-RNA-based overexpression (TRBO) and PVX-GFP has been tested. To determine the CasRx specificity, GFP sequence in TRBO–GFP has been replaced with a sequence encoding enhanced blue fluorescent protein (EBFP) and construct TRBO–BFP has been developed. CasRX targeting TRBO–BFP revealed the interference activity against the TRBO–BFP and not against the PVX-GFP. Similarly, when targeting PVX-GFP, only the GFP signal was mitigated and not the BFP signal produced by TRBO–BFP. No collateral activity was observed targeting a specific virus via CasRX.

The recent advancement and modification of the CRISPR/Cas system has enabled the development of a new precise base editing technique that modifies nucleotide (irreversible base conversion) rather than generating a double-stranded break [87,88]. This technique comprises the fusion of a nickase or nuclease-dead Cas9 with nucleotide deaminase guided by sgRNA bringing about targeted and precise mutagenesis. This method has been successfully applied for gene modification in several plants [89,90,91,92]. A recent study showed the precise use of base editing approach for the targeting and generation of point mutation into eukaryotic translational initiation factor (*eIF4E1* gene) of *A*. *thaliana,* and these point mutations were sufficient to restrict the accumulation of clover yellow vein virus without hampering plant growth [37].

## 4. Multiplex Editing

CRISPR/Cas9-derived mutations both inhibit geminiviral accumulation and generate viral escape [21,26]. One potential solution might be targeting two or more sites simultaneously in the viral genome with multiple sgRNAs. In the multiplex CRISPR approach, multiple sgRNAs or Cas nuclease are expressed simultaneously to edit or transcriptionally regulate various genetic loci in parallel [93]. By using the multiplex approach, either multiple genes of the same virus or numerous viruses are targeted at a time. The multiplexing of sgRNA against different coding/non-coding regions of the virus could evade the NHEJ repair system, the unrepaired molecules would be ultimately degraded [21]. The CRISPR/Cas9 approach against ACMV leads to the development of mutant virus (escape mutants), which prevents the development of resistance in cassava and that may drive the accelerated evolution of new viruses [26]. The mutant ACMV, having truncated AC2 ORF generated through CRISPR/Cas9, was able to replicate in *N. benthamiana* in the presence of wild-type ACMV, which can engage in the evolution of hypervirulent geminivirus isolates. Rybicki (2019) analyzed the findings of Mehta et al. (2019) and suggested that targeting several genes of viral genome through the multiplexing of sgRNA and proper expression of Cas9 could be an alternative approach to develop virus-resistant plants.

Roy et al. (2019) used the multiplex gRNA-based CRISPR/Cas9 approach targeting the geminivirus chilli leaf curl virus (ChiLCV) genome at two or more sites simultaneously. The strength of the constructs was tested transiently for the resistance of ChiLCV infection in *N. benthamiana*. The sgRNAs (2 sgRNA in combination) targeted C1/C4 and V1/V2 overlapping regions and IR of ChiLCV. The authors reported that C1/C4 and IR and IR/IR regions were more effective in lowering the virus titer and symptoms. A total of nine duplex (2 sgRNA in combination) and two triplex sgRNAs (3 sgRNA in combination) expressing constructs were developed and were agroinfiltrated into *N. benthamiana*. Virus titer was significantly reduced (60–90%) in the plants infiltrated with combinations of duplex/triplex gRNA compared with mock vector inoculated plants. A combination of two gRNAs targeting C1/C4 and V1/V2 showed the highest reduction (90%) in virus accumulation followed by C1/C4 and IR combination (85% reduction). A combination of three gRNA revealed a 70% reduction in virus titer. The appearance of disease symptoms was delayed 5–10 days in the systemic leaves of all multiplexed sgRNA-inoculated plants. The plants targeted with C1/C4 and V1/V2 gRNA-Cas9 construct did not show any leaf curl symptoms. The combinations of gRNA which showed the maximum reduction in virus accumulation were tested for the detection of mutation. T7E1 assay showed a distinct homoduplex band, confirming the presence of only wild-type virus population and failing to yield recognizable escape mutant fragment. Furthermore, sequencing analysis showed only the presence of wild-type virus. This study has shown the efficacy of multiplex targeting of the CRISPR-Cas system as compared with a single target.

## 5. Discussion, Conclusions and Future Directions

The SpCas9 protein has been used to target DNA viruses, while other nuclease proteins such as SpCas9, FnCas9, LshCas13a, RfxCas13d, LwaCas13a, PspCas13b and nCas9 have been used to target RNA viruses or its host genes. Among DNA viruses, targeting the Rep gene showed maximum reduction in virus accumulation that enhances the viral resistance in both transgenic and transient systems [18,19,22,25] (Table 1). Transgenic *A. thaliana* and *N. benthamiana* plants expressing Rep-sgRNA and IR-sgRNA, respectively showed higher resistance against BSCTV and the highest reduction in viral titers [19]. The efficiency of BSCTV inhibition is corelated with Cas9 expression level. Plants expressing multiple sgRNA (two or more sgRNA targeting viral genes simultaneously) showed either mild or no symptoms [23,24,25,29] (Table 1). Targeting essential viral proteins leads to the generation of escape mutants that replicate and move systemically in the plants [21,22,23,24,25,26]. The generation of escape mutants can be reduced through the multiplex sgRNA approach [29,39] and Cas9 expression under virus inducible promoter. In the case of plant viruses, targeting host factors (susceptibility factor), either by gene editing through Cas9 or by base editing through nCas9, showed complete resistance (no symptoms) and transgene-free mutant plants have been developed which could need less regulatory approvals and be more conducive to consumer acceptance than those bearing transgenes [34,35,36,37]. Other Cas proteins such as FnCas9 and LshCas13a (directly targeting RNA viruses), when used to target plant viruses, showed mild or no symptoms and a 50–99% reduction in virus accumulation [31,32,33] (Table 1).

If two major antiviral strategies, RNA silencing and genome editing are compared, genome editing provides novel methods for the improvement of virus resistance. RNA silencing is a highly efficient and successful approach and has generated commercialized transgenic antiviral crops, but many viruses encode viral suppressors of RNA silencing to counter the defense of RNA silencing. These suppressors are able to disrupt the RNAi-mediated silencing process by targeting key components of RNAi pathways [94]. Another disadvantage of using the RNAi antiviral system is that high mutation rate (10–20%) in homologous viruses can abolish plant resistance [95]. A genome editing strategy such as the CRISPR/Cas system can bypass these disadvantages. Other gene editing approaches such as zinc finger nuclease and TALEN are not widely used due to issues of affordability and efficiency. The rapid development of CRISPR technology is a definite milestone and has advantages over other systems. Plant viruses do not possess the ability to counter CRISPR-based immune defense originated from prokaryotes. Furthermore, by editing host factors and removing gene editing CRISPR machinery by backcross, virus-resistant transgene-free plants can be generated. Moreover, new CRISPR-based systems such as multiplexed CRISPR or CRISPR-Act2.0 can also be used to develop broad spectrum resistance against different kinds of pathogens by targeting multiple pathogens [93] or by transcriptional activation of plant dominant resistance gene(s) [96], respectively. Nevertheless, these new systems also have limitations such as off targeting or production of escape virus mutants, but in a longer horizon, these problems can be solved (Figure 2).

The efficacy of CRISPR-based approaches primarily depends on delivery method and sgRNA sequence. A number of delivery methods including *Agrobacterium*-mediated T-DNA transformation, protoplast transfection, microprojectile bombardment and virus-based methods have been deployed to introduce CRISPR/Cas components in plants. The system most commonly employed to obtain transgenic plants is based on *Agrobacterium tumefaciens* and has been used to deliver the CRISPR/Cas system in varieties of plant species. Cas9 and sgRNA expressing genes are cloned into Ti plasmid and then introduced in plants. The stable integration of transgenes into many plant genomes such as *A. thaliana*, rice, tomato, maize, grapevine and sorghum has been achieved successfully with editing efficiencies from 23% to 100% [97,98,99,100,101].

The stability of CRISPR-edited virus resistance plants in subsequent generation has also been demonstrated. In the case of DNA viruses, Tashkandi et al. (2018) demonstrated that T_3_ tomato plants having sgRNA and Cas9 showed low accumulation of viral genome. In the case of RNA virus modification of host factors such as eIf(iso)4E, eIF4E and eIF4G, it showed complete resistance against TuMV, CVYV and RTSV in T_3_ Arabidopsis, T_3_ cucumber and T_2_ rice plants, respectively [34,35,36]. Inheritability of CMV resistance in T_6_ generation Arabidopsis having FnCas9 and sgRNA was tested, and T_6_ Arabidopsis plants showed stable resistance to CMV [31]. Although CRISPR-edited virus-resistant plants have not been released in soil so far, CRISPR-edited crops such as high amylopectin waxy corn (*Zea mays*), false flax (*Camelina sativa*) and browning-resistant mushrooms have been approved by the U.S. Department of Agriculture, and false flax with enhanced omega-3 oil is expected to reach the U.S. market [102,103]. Recently, CRISPR-edited Sicilian Rouge tomatoes congaing a high amount of GABA (γ-aminobutyric acid) have been released in Japan [104]. This suggests that CRISPR-edited virus-resistant crops will reach the market and will be available for growers in the near future.

The CRISPR/Cas9 system has become a vital tool for gene editing technology. It employs sequence-specific endonuclease-based technology composed of two components: Cas9 nuclease and an engineered sgRNA targeting any DNA sequence. The last decade reported multiple approaches on the development of the CRISPR/Cas-mediated virus resistance system in different crops and model plants. To target DNA virus, single or multiple viral effectors can be targeted in the viral genome itself. Moreover, targeting the IR common region and Ori also resulted in broad spectrum resistance against viruses. The approach to target RNA viruses is different. The CRISPR/SpCas9 or CRISPR/dCas9 system has been developed against host factors such as eIF4E, which RNA viruses require to replicate their genomes. Since this knocking out will result in non-functional protein, developed resistance is highly durable. CRISPR/Cas9 has limitations in editing efficiency due to hindrance because of its large size. To overcome this, a variety of CRISPR systems have been generated for efficient gene editing, including the SpCas9 system.

Although these novel CRISPR/Cas system-based developments of viral-resistant plants are effective, efficient and revolutionary, crop improvement still requires more durability and broad-spectrum resistance. Moreover, minimizing multiple off-target effects due to this technology is an immediate concern. Finally, in order to prevent the emergence of CRISPR/Cas escape mutant viruses, multiple combinatorial antiviral approaches can be employed. The identification of novel host factors that are essential for viral propagation and their targeting, using this promising approach, will drive this technology further.

## Figures and Tables

**Figure 1 ijms-23-02303-f001:**
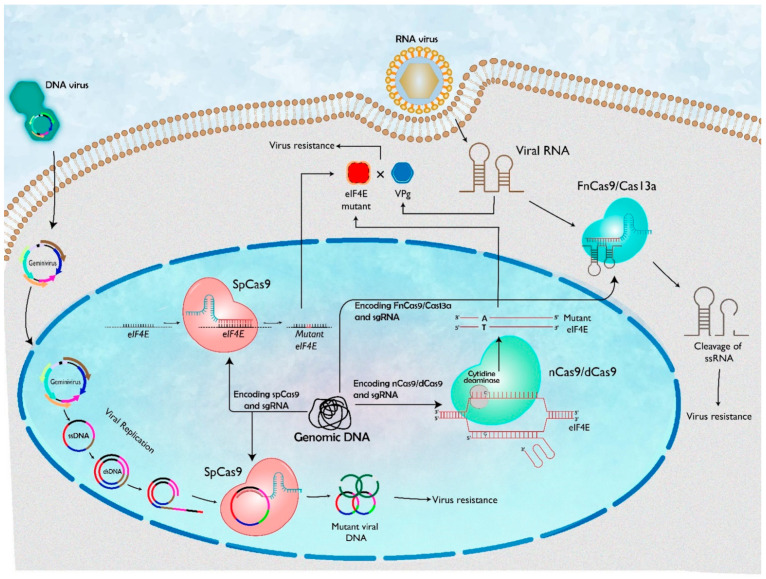
Representation of CRISPR/Cas systems to confer resistance against DNA and RNA virus. On entering plant cells, the viral genome is transcribed and translated using host factors. Subsequently, the viral genome is multiplied and spread to other cells. The viral genome can be targeted by the CRISPR system using Cas9/nCas9/dCas9/FnCas9/Cas13a as endonuclease. Components of the CRISPR/Cas9 machinery, sgRNA and Cas9 are expressed from the plant genome and form sgRNA-Cas9 complex. Upon viral infection, the viral DNA replicates through the dsDNA replicative form inside the nucleus of the host cell. The sgRNA-Cas9 complex targets the viral dsDNA and cleaves or mutates the viral genome. Similarly, viral RNA is targeted by sgRNA-FnCas9/Cas13a generating mutation in RNA. Potyviruses recruit host cellular translation factor eIF4E to translate their viral RNA and facilitate their infection. Thus, eukaryotic translation initiation factors eIF4E can be used as a target for CRISPR/Cas9, resulting in a recessive gene mutant form. This mutant form of eIF4E is not able to interact with VPg, a viral translation machinery protein. Furthermore, nuclease-defective Cas (dCas9/nCas9) employs base editing to impair viral RNA. This system contains a fused cytidine deaminase and a fused uracil glycosylase inhibitor (UGI).

**Figure 2 ijms-23-02303-f002:**
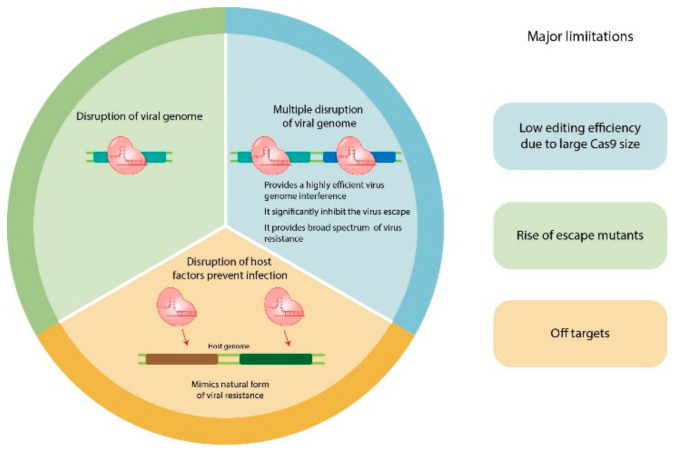
Overview of strategies to combat plant viruses using the CRISPR system. There are three general strategies, and they are discussed in this review to target viral genomes in infected cells. The top left panel displays the directing of genome of DNA or RNA viruses using single sgRNAs by which essential viral genes can be disrupted or the viral genome can be fragmented. The top right panel describes strategies that can be used to target multiple genes of the same or different virus. The lower panel displays the potential to target host factors that are essential for viral propagation and maintenance.

**Table 1 ijms-23-02303-t001:** Application of CRISPR/Cas in the development of virus resistance in plants.

Virus Name	Genus; Family	Effector Protein	Delivery of Constructs	Target Region	Resistance Levels	Host Plant	% Reduction in Viral Titers	% Mutations of the Target	Refs.
Direct target on DNA viruses									
Bean yellow dwarf virus	*Mastervirus; Geminiviridae*	SpCas9	*Agrobacterium tumefaciens*; sgRNA/Cas9; transgenic plants	LIR	Mild symptoms	*Nicotiana benthamiana*	71	70	[18]
				Rep/RepA	Mild symptoms	*N. benthamiana*	78	20	
Beet severe curly top virus	*Curtovirus*; *Geminiviridae*	SpCas9	*A. tumefaciens*; sgRNA/Cas9; transgenic plants	IR	Mild or no symptoms	*N. benthamiana*	30–90	NA	[19]
				CP	Mild symptoms	*N. benthamiana*	20–90	NA	
				Rep	No symptoms	*N. benthamiana* *Arabidopsis thaliana*	70–95	NA	
Tomato yellow leaf curl virus (TYLCV)	*Begomovirus*; *Geminiviridae*	SpCas9	*A. tumefaciens* and TRV-mediated delivery of sgRNA in Cas9 expressing transgenic plants	IR	Mild symptoms or no symptoms	*N. benthamiana*	NA	36–42	[20]
				CP	Severe symptoms	*N. benthamiana*	NA	22–28	
				Rep	Severe symptoms	*N. benthamiana*	NA	31–39	
Cotton leaf curl Kokhran virus, Merremia mosaic virus	*Begomovirus*; *Geminiviridae*	SpCas9	*A. tumefaciens* for overexpression of Cas9 and TRV-mediated delivery of sgRNA	CP	NA	*N. benthamiana*	NA	18–49	[21]
				Rep	NA	*N. benthamiana*	NA	35–45	
				IR	NA	*N. benthamiana*	NA	NA	
Tomato yellow leaf curl virus	*Begomovirus*; *Geminiviridae*	SpCas9	*A. tumefaciens*; sgRNA/Cas9; transgenic plants	CP	NA	*Solanum lycopersicum*	69	34–69	[22]
				Rep	NA	*S. lycopersicum*	74	14–37	
				IR	NA	*S. lycopersicum*	NA	NA	
Cotton leaf curl Multan virus	*Begomovirus*; *Geminiviridae*	SpCas9	*A. tumefaciens;* sgRNA/Cas9; transgenic plants	Rep + IR	No symptoms	*N. benthamiana*	NA	NA	[23]
Wheat dwarf virus	*Mastrevirus*; *Geminiviridae*	SpCas9	*A. tumefaciens;* sgRNA/Cas9; transgenic plants	MP/CP + Rep/RepA + LIR	No symptoms	Barley (*Hordeum vulgare* L.)	NA	NA	[24]
Cotton leaf curl virus and betasatellite	*Begomovirus*; *Geminiviridae*	SpCas9	*A. tumefaciens;* sgRNA/Cas9; transient expression	Rep	Mild (delayed symptoms (2–4 days))	*N. benthamiana*	40–70	NA	[25]
				Rep + βC1	Mild (delayed symptoms (3–5 days))	*N. benthamiana*	60–80	NA	
African cassava mosaic virus	*Begomovirus*; *Geminiviridae*	SpCas9	*A. tumefaciens;* sgRNA/Cas9; transgenic plants	AC2/AC3 (TrAp/REn)	Mild to severe symptoms	Cassava	0	11	[26]
Cauliflower mosaic virus	*Caulimovirus*; *Caulimoviridae*	SpCas9	*A. tumefaciens;* sgRNA/Cas9; transgenic plants	CP	15% plants showed symptoms	*A. thaliana*	20–52	43	[27]
Banana streak virus	*Badnavirus*; *Caulimoviridae*	SpCas9	*A. tumefaciens;* sgRNA/Cas9; transgenic plants	ORF1, 2, 3	25% showed symptoms	Banana (*Musa* spp.)		70–85	[28]
Chilli leaf curl virus	*Begomovirus*; *Geminiviridae*	SpCas9	*A. tumefaciens;* sgRNA/Cas9; transient expression	C1/C4 + V1/V2	No symptoms	*N. benthamiana*	90	NA	[29]
				C1/C4 + IR	No symptoms	*N. benthamiana*	85	NA	
				C1/C4 + V1/V2 + IR	Mild symptoms	*N. benthamiana*	70	NA	
Host modification									
TYLCV	*Begomovirus; Geminiviridae*	SpCas9	*A. tumefaciens;* sgRNA/Cas9; transgenic plants	SlPelo	No symptoms	*Solanum lycopersicum*	90–100	10	[30]
Direct target on RNA viruses									
Cucumber mosaic virus	*Cucumovirus; Bromoviridae*	FnCas9	*A. tumefaciens;* sgRNA/Cas9; transient expression	ORF1a, ORF 3a, 3′UTR	Mild symptoms	*N. benthamiana*	50–60	NA	[31]
			*A. tumefaciens;* sgRNA/Cas9; transgenic plants	ORF1a, ORF 3a, 3′UTR	No symptoms	*A. thaliana*	70–85	NA	
Potato virus Y	*Potyvirus; Potyviridae*	LshCas13a	*A. tumefaciens;* sgRNA/Cas13a; transgenic plants	P3, CI, Nib, CP	No symptoms	Potato (*Solanum tuberosum*)	99	NA	[32]
TuMV	*Potyvirus*; (*Potyviridae*)	LshCas13a	*A. tumefaciens;* sgRNA/Cas13a; transient expression	Different location in TMV genome		*N. benthamiana*	70–80	NA	[33]
Southern rice black-streaked dwarf virus (SRBSDV), Rice Stripe Mosaic Virus (RSMV)	*Fijivirus* (*Reoviridae*), *Cytorhabdovirus* (*Rhabdoviridae*)	LshCas13a	*A. tumefaciens;* sgRNA/Cas13a; transgenic plants	Various locations in SRBSDV and RSMV genome	Mild symptoms	Rice	60–80	NA	[33]
Host modification									
Turnip mosaic virus (TuMV)	*Potyvirus*; *Potyviridae*	SpCas9	*A. tumefaciens;* sgRNA/Cas9; transgenic plants	eIF(iso)4E	No symptoms	*A. thaliana*	100	NA	[34]
Cucumber vein yellowing virus, Zucchini yellow mosaic virus, Papaya ring spot mosaic virus-W	*Ipomovirus;* *Potyvirus; Potyvirus; Potyviridae*	SpCas9	*A. tumefaciens;* sgRNA/Cas9; transgenic plants	eIF4E	No symptoms (Homozygous mutant), Severe symptoms (heterozygous mutant)	Cucumber (*Cucumis sativus* L.)	100	NA	[35]
Rice tungro spherical virus	*Waikavirus; Secoviridae*	SpCas9	*A. tumefaciens;* sgRNA/Cas9; transgenic plants	eIF4G	Resistant	Rice (*Oryza sativa* var. indica cv. IR64)	NA	59	[36]
Clover yellow vein virus	*Potyvirus; Potyviridae*	nCas9	*A. tumefaciens;* sgRNA/Cas9; transgenic plants	eIF4E1	Complete resistance	*A. thaliana*	NA	31	[37]

## Data Availability

Not applicable.

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
