# Peer review of "CRISPR/Cas-Mediated Resistance against Viruses in Plants"

_ijms, 2022, doi:10.3390/ijms23042303_

Round 1

Reviewer 1 Report

Dear Editor,

Thank you for inviting me to review the article titled with " CRISPR/Cas-mediated resistance against viruses in plants" I see that the idea of the article is sound good and the authors used many cases of study to  reach their point of view. But authors did not remind to us with which crop the CRISPR/CASE was success and resulted in a resistant plant for specific virus.  Also, there is another genes could be used and success more than Cas, such as viral replication gene, viral movement gene, viral assembling gene and etc. The authors did not give us a right way on which the scientist could use it to make transgenic plant using the  CRISPR/CASE system and the main purpose of them" including multiplexing to increase editing efficiency and bypassing the generation of escape mutants" This two points editing efficacy and escaping mutants should be focused in and discussed in good manner. The authors draw only one figure for what previously studied and I see that they need to draw another figure showing their idea and the future expectation to it.  I see that authors should compare the  CRISPR/CASE system and the other systems used in developing transgenic plants and indicate the percentage of success for the both and which more the best. I need to ask the authors what about the genetic stability of the  CRISPR/CASE system, is the stability of this system was studied or not? and what happened if this system was released into the soil.

Author Response

Reviewer 1

Dear Editor,

Thank you for inviting me to review the article titled with "CRISPR/Cas-mediated resistance against viruses in plants" I see that the idea of the article is sound good and the authors used many cases of study to reach their point of view. But authors did not remind to us with which crop the CRISPR/CASE was success and resulted in a resistant plant for specific virus.  Also, there is another genes could be used and success more than Cas, such as viral replication gene, viral movement gene, viral assembling gene and etc. The authors did not give us a right way on which the scientist could use it to make transgenic plant using the CRISPR/CASE system and the main purpose of them" including multiplexing to increase editing efficiency and bypassing the generation of escape mutants" This two points editing efficacy and escaping mutants should be focused in and discussed in good manner. The authors draw only one figure for what previously studied and I see that they need to draw another figure showing their idea and the future expectation to it.  I see that authors should compare the  CRISPR/CASE system and the other systems used in developing transgenic plants and indicate the percentage of success for the both and which more the best. I need to ask the authors what about the genetic stability of the CRISPR/CASE system, is the stability of this system was studied or not? and what happened if this system was released into the soil.

Reply: We have added another figure as recommended

Figure 2. Overview of strategies to combat plant viruses using CRISPR system. There are three general strategies and are discussed in this review to target viral genomes in infected cells. The top left panel displays directing of genome of DNA or RNA viruses using single sgRNAs by which essential viral genes can be disrupted or the viral genome can be fragmented. The top right panel describes strategies that can be used to target multiple genes of same or different virus. Lower panel displays potential to target host factors that are essential for viral propagation and maintenance.

Reply: We have added these points in discussion section

If two major antiviral strategies, RNA silencing and genome editing are compared, genome editing provides novel methods for the improvement of virus resistance. Despite RNA silencing being highly efficient and successful approach and capable of generating commercialized transgenic virus-resistant crops but many viruses can encode viral suppressors of RNA silencing to counter the defense of RNA silencing. These suppressors are able to disrupt RNAi mediated silencing process by targeting key components of RNAi pathways [95]. Other disadvantage of using RNAi antiviral system is that high mutation rate (10-20%) in homologous viruses can abolish the plant resistance [96]. Genome editing strategy such as CRISPR/Cas system bypass these disadvantages. Other gene editing approaches, for example zinc finger nuclease and TALEN are not widely used due to issues of affordability and efficiency. The rapid development of CRISPR technology is a definite milestone and has advantages over other systems as per counter defense by viruses. Plant viruses do not possess the ability to counter such immune defense originated from prokaryotes. Furthermore, by editing host factors and removing gene editing CRISPR machinery by backcross, virus resistant transgene free plants can be generated. Moreover, new CRISPR based systems like multiplexed CRISPR or CRISPR-Act2.0 can also be used to develop broad spectrum resistance against different kind of pathogens by targeting multiple pathogens [94] or by transcriptional activation of plant dominant resistance gene(s) [97], respectively. Nevertheless, these new systems also have limitations such as off targeting or production of escape virus mutants, but in a longer horizon, these problems can be solved (Figure 2).

Efficacy of CRISPR based approaches primarily depends on delivery method and sgRNA sequence. Number of delivery methods including Agrobacterium-mediated T-DNA transformation, protoplast transfection, microprojectile bombardment and virus-based methods have been deployed to introduce CRISPR/Cas components in plants. The system which is most commonly used to obtain transgenic plants is based on Agrobacterium tumefaciens and has been used to deliver CRISPR/Cas system in varieties of plant species. Cas9 and sgRNA expressing genes are cloned into Ti plasmid and then it is introduced in plants. Stable integration of transgenes into many plant genomes such as A. thaliana, rice, tomato, maize, grapevine and sorghum have been achieved successfully with editing efficiencies from 23% to 100% [98-102].

We discussed the stability of CRISPR-edited virus resistance plants in subsequent generation. In the case of DNA viruses, Tashkandi et al. (2018) demonstrated that T3 tomato plants having sgRNA and Cas9 showed low accumulation of viral genome. In the case of RNA viruses, modification of host factors such as eIf(iso)4E, eIF4E, eIF4G showed complete resistance against TuMV, CVYV and RTSV in T3 Arabidopsis, T3 cucumber and T2 rice plants, respectively [19,20,34]. Inheritability of CMV resistance in T6 Arabidopsis having FnCas9 and sgRNA was tested and T6 Arabidopsis plants showed stable resistance to CMV [35]. Although, CRISPR-edited virus resistance plants has not released in soil so far, but, CRISPR-edited crops such as high amylopectin waxy corn (Zea mays), false flax (Camelina sativa) and browning resistant mushrooms have been approved by the US Department of Agriculture and false flax with enhanced omega-3 oil is expected to reach the US market [103,104]. Recently, CRISPR-edited Sicilian Rouge tomatoes containing high amount of GABA (γ-aminobutyric acid) have been released in Japan [105]. This suggests that CRISPR-edited virus resistant crops will reach in the market and will be available for the growers in near future. 

Reviewer 2 Report

The review manuscript submitted by Zainul A. Khan et al. summarized how to improve virus resistance in plants by using CRISPR/Cas systems. There are two approaches, one is targeting virus itself, another is CRISPR/Cas systems-mediated host genome modification. Overall, the described topic is interesting and will contribute to the molecular breeding. In the meantime, there are few aspects that fall short as detailed below, these need to be addressed.

Table 1. seems too complicated for me. How did the authors arrange these publications? I cannot find out any rules. I would like to recommend to rearrange the Table 1. For example, sort by virus name and category, which means DNA or RNA virus. Another issue is ‘delivery of constructs’. Most of them were delivered by Agrobacterium, but some were for stable stransformant, others were transient infiltration. The authors should clarify the difference of delivery methods.

Page 7, line218-220. Is this correct? Are there any specific mechanisms to silence transgenes, which is generated for virus-resistance, in plants? In my understanding, the cited manuscript (Daxinger et al., 2008) did not describe such kind of machinery.

Page 9, line 323-343. What do the G0 and G1 mean? The authors did not use these terms in the original manuscript (Lapidot et al., 2015).

Page 10, line 403-410. Could you explain more details what the hemizygous mean, one nucleotide deletion and 20 nt deletion? In my understanding, this cucumber is two times backcrossing plant. If so, one allele is CRISPR/Cas-mediated mutation, another allele comes from Bet Alfa, which must be wild-type. I am wondering how to obtain hemizygous plant. Further, transgene free plant was selected at the first time backcrossing. It is not clear for me how to exist two different kinds of mutations in the next generation, and how to generate new mutation without CRISPR/Cas system. Moreover, I think ‘biallelic’ should be better choice instead of hemizygous, which usually means one allele exists, but not another allele.

The authors described specificity of sgRNA in several times in the manuscript. I think the most important information for readers is how much is the sequence identity between possible sgRNA target sites among virus (line 501, 535, and so on). I would like to recommend the authors to clarify the similarities.

Page 14, line 572-581. Multiple sgRNAs showed higher resistance compare with mock vector. I am wondering how is the efficiency of single sgRNA. It should be important how much resistance increased in multiple sgRNA systems than single sgRNA.

Minor comments

Figure 1. spCas9 should be SpCas9. Escape mutant of virus should also be represented.

Page 5, line 102, 103, 107, 118, 117, 122, 125, and so on. The name of virus family must be italic.

Page 5, 124. lea fhoppers should be leaf hoppers.

Page 9, line 321-323. The citation seems incorrect. Pramanik et al., is 22.

Page 10, line 367. The size of eIF4E and eIF(iso)4E should be similar. The size of proteins must be corrected (180 and 86 KDa).

Page 11, line421. CRISPR-Cas. The writing of terms should be uniformed.

Author Response

Reviewer 2

The review manuscript submitted by Zainul A. Khan et al. summarized how to improve virus resistance in plants by using CRISPR/Cas systems. There are two approaches, one is targeting virus itself, another is CRISPR/Cas systems-mediated host genome modification. Overall, the described topic is interesting and will contribute to the molecular breeding. In the meantime, there are few aspects that fall short as detailed below, these need to be addressed.

Table 1. seems too complicated for me. How did the authors arrange these publications? I cannot find out any rules. I would like to recommend to rearrange the Table 1. For example, sort by virus name and category, which means DNA or RNA virus. Another issue is ‘delivery of constructs’. Most of them were delivered by Agrobacterium, but some were for stable transformants, others were transient infiltration. The authors should clarify the difference of delivery methods.

Reply: Publications are arranged as per the published year of the research paper which we discussed in this MS. Rearranged Table 1 and viruses are sorted by category, first DNA viruses and then RNA viruses, as suggested. In the ‘delivery of Constructs’ section, now, we have written which one are transgenic and which one are transient, that clearly differentiate between stable and transient expression of desired gene delivered through Agrobacterium. In most of the cases sgRNA and Cas are delivered in plants through Agrobacterium. 

Page 7, line 218-220. Is this correct? Are there any specific mechanisms to silence transgenes, which is generated for virus-resistance, in plants? In my understanding, the cited manuscript (Daxinger et al., 2008) did not describe such kind of machinery.

Reply: We corrected the sentence… “As the silencing of the T-DNA inserted genes (transgene) happen in plants [54], Tashkandi et al. (2018) tested virus interference through multiple generations”.

Earlier study showed that silencing of the T-DNA harboring genes can occur in successive generations (Daxinger et al., 2008).

Page 9, line 323-343. What do the G0 and G1 mean? The authors did not use these terms in the original manuscript (Lapidot et al., 2015).

Reply: G0 means genome-edited generation 0 and G1 is genome-edited generation 1.

Pramanik et al., 2021 used these terms in their manuscript which we discussed in this section.

L 338-341: We modified the sentence “Similarly, when Pramanik et al. (2021) challenged CRISPR based SlPelo-edited knockout G1 plants against TYLCV, they reported no symptoms in TYLCV infected plants and significant reduction in viral load.”

Page 10, line 403-410. Could you explain more details what the hemizygous mean, one nucleotide deletion and 20 nt deletion? In my understanding, this cucumber is two times backcrossing plant. If so, one allele is CRISPR/Cas-mediated mutation, another allele comes from Bet Alfa, which must be wild-type. I am wondering how to obtain hemizygous plant. Further, transgene free plant was selected at the first time backcrossing. It is not clear for me how to exist two different kinds of mutations in the next generation, and how to generate new mutation without CRISPR/Cas system. Moreover, I think ‘biallelic’ should be better choice instead of hemizygous, which usually means one allele exists, but not another allele.

Reply: Line 405-415; Rewritten as follows:

Sequence analysis of the cloned target region of To transgenic plants having sgRNA1 showed two types of mutations, a 20 nucleotide deletion around the PAM and 1 nucleotide deletion at 3 bp upstream of PAM. T0 mutant plant (derived from ‘Ilan’, a multi-pistillate, parthenocarpic glasshouse cucumber) was cross-pollinated with ‘Bet Alfa’ (a monoecious, non-parthenocarpic, open field cucumber) for propagation by seeds. As expected for a single transgene locus, the segregation of transgenic to non-transgenic in the T1 generation was observed approximately 1:1. Four types of plants were appeared in T1 generation, a 20 nt deletion (non-transgenic, lacking of Cas9), one nt deletion (transgenic), one nt deletion (non-transgenic, lacking of Cas9) and both one and 20 nt deletion (non-transgenic, lacking of Cas9). The non-transgenic T1 plants having both one nt and 20 nt deletion was cross-pollinated once again with the monoecious ‘Bet Alfa’. The T2 progeny representing hemizygous for one nucleotide deletion and 20 nt deletion were self-pollinated for the generation of T3 lines. In T3 population, 20 nt deletion segregated in Mendelian manner in 1:2:1 ratio (homozygous : heterozygous : wild-type without mutation). T3 non-transgenic homozygous, heterozygous and wild-type (non-mutant) plants were inoculated with the cucumber vein yellowing virus (CVYV, genus Ipomovirus), zucchini yellow mosaic virus (ZYMV, genus Potyvirus), papaya ring spot mosaic virus-W (PRSV-W, Potyvirus), cucumber mosaic virus (CMV, genus Cucumovirus) and cucumber green mottle mosaic (CGMMV, genus Tobamovirus).

The authors described specificity of sgRNA in several times in the manuscript. I think the most important information for readers is how much is the sequence identity between possible sgRNA target sites among virus (line 501, 535, and so on). I would like to recommend the authors to clarify the similarities.

Reply: Line 501. When compared respective four sgRNA sequences with genomic DNA of PVA and PVS, it was found spacer sequences to be less than 35% similar with their respective targets.

Species demarcation criteria among potyviruses is <76% nucleotide sequence identity. While the genus demarcation criteria is <46% nt identity. Potato virus Y (PVY) and Potato virus A (PVA) belong to genus Potyvirus, while Potato virus S (PVS) comes under genus Carlavirus.

Page 14, line 572-581. Multiple sgRNAs showed higher resistance compare with mock vector. I am wondering how is the efficiency of single sgRNA. It should be important how much resistance increased in multiple sgRNA systems than single sgRNA.

Reply: We agree, multiplexed sgRNAs should be compared with the single sgRNA. The authors (Roy et al., 2019) did not compare the efficacy of single sgRNA with multiplexed sgRNAs.

Minor comments

Figure 1. spCas9 should be SpCas9. Escape mutant of virus should also be represented.

Reply: Corrected SpCas9

Page 5, line 102, 103, 107, 118, 117, 122, 125, and so on. The name of virus family must be italic.

Reply: Names of the virus families are italicized as suggested

Page 5, 124. lea fhoppers should be leaf hoppers.

Reply:  leafhoppers is correct

Page 9, line 321-323. The citation seems incorrect. Pramanik et al., is 22.

Reply: Cui et al., 2019 [65] is cited here for how pairs of sgRNA (multiplexing) targeting single gene using CRIPR/Cas9 produced large deletion in wheat. It means multiplexing of single gene increases large deletions and knockout generation. Cui et al., (2019) showed that co-expression of pairs of sgRNA targeting single genes in conjunction with the CRISPR/Cas9 system produced large deletions in wheat.

L 321-323: We modified the sentence “Pramanik et al. (2021) designed plasmid vector carrying sgRNAs for editing of single gene which increase chance of large deletions and knockout generation as previously reported [65].

Page 10, line 367. The size of eIF4E and eIF(iso)4E should be similar. The size of proteins must be corrected (180 and 86 KDa).

Reply: eIF4E1 and eIF(iso)4E shared 41% amino acid sequence identity and both have similar molecular mass of approximately 24 kDa. While, eIF4G and eIF(iso)4G have 27% amino acid identity and have differ molecular mass 168 kDa and 86 kDa, respectively.

Line 365-367 rewritten as follows:

  1. thaliana encodes three isoforms of the eIF4F/eIF(iso)4F large subunits, eIF4G, eIF(iso)4G1, and eIF(iso)4G2 and four isoforms of the small subunit cap-binding protein, eIF4E1, eIF4E2, eIF4E3, and eIF(iso)4E. eIF4E1 and eIF(iso)4E share 41% amino acid sequence identity and both have similar molecular mass of approximately 24 kDa. However, eIF(iso)4G isoforms differ in molecular mass from the eIF4G (83/86 kDa Vs. 168 kDa, respectively) and have 27% amino acid identity to eIF4G (Lellis et al., 2010)[71].  

Page 11, line421. CRISPR-Cas. The writing of terms should be uniformed.

Reply: Corrected CRISPR/Cas

References added

  1. Qu, F. Plant viruses versus RNAi: simple pathogens reveal complex insights on plant antimicrobial defense. Wiley Interdiscip. Rev. RNA 2010, 1, 22-33.
  2. Kumar, A.; Sarin, N.B. RNAi: A promising approach to develop transgenic plants against geminiviruses and insects. J. Plant. Physiol. Pathol. 2013, 1, 1.
  3. Lowder, L.G.; Zhou, J.; Zhang, Y.; Malzahn, A.; Zhong, Z.; Hsieh, T.F.; Voytas, D.F.; Zhang, Y.; Qi, Y. Robust Transcriptional activation in plants using multiplexed CRISPR-Act2.0 and mTALE-Act systems. Mol. Plant. 2018, 11, 245-256.
  4. Miao, J.; Guo, D.; Zhang, J.; Huang, Q.; Qin, G.; Zhang, X.;Wan, J.; Gu, H.; Qu, L.J. Targeted mutagenesis in rice using CRISPR-Cas system. Cell Res. 2013, 23, 1233-1236.
  5. Feng, Z.; Mao, Y.; Xu, N.; Zhang, B.;Wei, P.; Yang, D.L.;Wang, Z.; Zhang, Z.; Zheng, R.; Yang, L.; et al. Multigeneration analysis reveals the inheritance, specificity, and patterns of CRISPR/Cas-induced gene modifications in Arabidopsis. Proc. Natl. Acad. Sci. USA 2014, 111, 4632-4637.
  6. Pan, C.; Ye, L.; Qin, L.; Liu, X.; He, Y.; Wang, J.; Chen, L.; Lu, G. CRISPR/Cas9-mediated efficient and heritable targeted mutagenesis in tomato plants in the first and later generations. Sci. Rep. 2016, 6, 24765.
  7. Char, S.N.; Neelakandan, A.K.; Nahampun, H.; Frame, B.; Main, M.; Spalding, M.H.; Becraft, P.W.; Meyers, B.C.;Walbot, V.;Wang, K.; et al. An Agrobacterium-delivered CRISPR/Cas9 system for high-frequency targeted mutagenesis in maize. Plant Biotechnol. J. 2017, 15, 257-268.
  8. Nakajima, I.; Ban, Y.; Azuma, A.; Onoue, N.; Moriguchi, T.; Yamamoto, T.; Toki, S.; Endo, M. CRISPR/Cas9-mediated targeted mutagenesis in grape. PLoS ONE 2017, 12, e0177966.
  9. Waltz, E. CRISPR-edited crops free to enter market, skip regulation. Nat. Biotechnol. 2016, 34, 582.
  10. Waltz, E. With a free pass, CRISPR-edited plants reach market in record time. Nat. Biotechnol. 2018, 36, 6-7.
  11. Waltz, E. GABA-enriched tomato is first CRISPR-edited food to enter market. Nat. Biotechnol. 2022, 40, 9-11.

Round 2

Reviewer 2 Report

The authors have dealt with the critical points which I raised in the original submission in an adequate way. The revised version of manuscript is improved.